# Vertical Vibration of Mouse Osteoblasts Promotes Cellular Differentiation and Cell Cycle Progression and Induces Aging In Vitro

**DOI:** 10.3390/biomedicines11020444

**Published:** 2023-02-03

**Authors:** Daehwan Choi, Takenobu Ishii, Munetada Ishikawa, Tomohisa Ootake, Hirokazu Kamei, Kohei Nagai, Kenji Sueishi

**Affiliations:** 1Department of Orthodontics, Tokyo Dental College, 2-9-18, KandaMisaki-Cho, Chiyoda-ku, Tokyo 101-0061, Japan; dh.c.ortho@gmail.com (D.C.); ortho@chiba-kyousei.jp (M.I.); ootaketomohisa7@gmail.com (T.O.); kameihirokazu@tdc.ac.jp (H.K.); knagai52@gmail.com (K.N.); sueishi@tdc.ac.jp (K.S.); 2Department of Orthodontics, Tokyo Dental College Chiba Dental Center, 1-2-2, Masago, Mihama-ku, Chiba 261-0011, Japan

**Keywords:** vibration, osteoblast, aging, cell cycle, cell proliferation

## Abstract

Background: This study aimed to investigate the effect of the vibration of osteoblasts on the cell cycle, cell differentiation, and aging. Materials and Methods: Primary maxilla osteoblasts harvested from eight-week-old mice were subjected to vibration at 3, 30, and 300 Hz once daily for 30 min; control group, 0 Hz. A cell proliferation assay and Cell-Clock Cell Cycle Assay were performed 24 h after vibration. Osteoblast differentiation assay, aging marker genes, SA-β-Gal activity, and telomere length (qPCR) were assayed two weeks post- vibration once every two days. Results: Cell proliferation increased significantly at 30 and 300 Hz rather than 0 Hz. Several cells were in the late G_2_/M stage of the cell cycle at 30 Hz. The osteoblast differentiation assay was significantly higher at 30 Hz than at 0 Hz. Runx2 mRNA was downregulated at 30 Hz compared to that at 0 Hz, while osteopontin, osteocalcin, and sclerostin mRNA were upregulated. p53/p21, p16, and c-fos were activated at 30 Hz. SA-β-Gal activity increased significantly at 30 or 300 Hz. Telomere length was significantly lower at 30 or 300 Hz. Conclusions: The results suggest that providing optimal vibration to osteoblasts promotes cell cycle progression and differentiation and induces cell aging.

## 1. Introduction

Application of vibration to bones reportedly influences bone metabolism. Studies have reported improvements in healing of fractured bones after application of appropriate, perpendicular micro-vibrations to the tibia during the initial stages of bone fracture [1]; studies have also reported increased bone quality and strength in sheep femurs after systemic application of vibration [2,3], increased bone density after low-intensity vibrations to rat and mouse bones [4,5], and improved bone density in the spinal cord and tibia of pediatric patients after vibration treatment [6]. Moreover, studies have reported a significant increase in alkaline phosphatase (ALP) activity [7] and calcium deposition [8] after vibration. Various studies have investigated the biological effects of low-frequency (0.5 Hz) to ultrasonic vibrations (several MHz). Meanwhile, the application of vibration at 30–130 Hz is currently being attempted to increase the efficiency of orthodontic treatment. Mechanisms of tooth movement are composed of osteogenesis on the root tension side and osteoclastogenesis on the root compression side. Studies have utilized tooth movement models in rats [9] and clinical trials [10]. While such studies reported that application of vibration during tooth retraction accelerated tooth movement, the effects and underlying mechanisms are still debated upon [11]. Since the effects have not been clarified in the orthodontic treatment, the application of vibration to orthodontic tooth movement has not been established.

The application of vibration to osteoblasts reportedly improves bone quality and osteoblast gene expression and induces alterations during calcification [1,2,3,4,5,6,7,8]. However, few studies have investigated the underlying cellular phenomena causing these changes. In the present study, we hypothesized that these effects result from the acceleration of the cell cycle. Therefore, we speculated that the application of vibration would accelerate the cell cycle and promote cell proliferation and differentiation. Furthermore, we hypothesized that cell cycle acceleration induces cellular aging. Accordingly, this study aimed to investigate the effect of vibration on cultured primary osteoblasts derived from mouse maxillary bone for a specified period on cell cycle progression, cellular differentiation, and cell aging.

## 2. Materials and Methods

### 2.1. Cell Culture

Primary osteoblasts (POB cells) were isolated from the maxillary bone of 8-week-old male mice (C57BL/6J, Charles River, Wilmington, MA, USA). The maxillary bone of C56BL/6J mice was obtained, and the first to third molars were extracted using an injection needle under a microscope. After removal of the soft tissue, the maxillary bone was treated with PBS containing 0.1% collagenase (Wako Pure Chemical, Osaka, Japan) and 0.2% Disperse II (Godo Shusei, Chiba, Japan). The digestion procedure was repeated three times, and the cells collected here were cultured as osteoblasts. The POB cells were cultured at 37 °C and 5% CO2 in alpha-minimum essential medium (α-MEM; Gibco, Grand Island, NY, USA) media supplemented with 10% fetal bovine serum (FBS; Gibco), penicillin G (100 U/mL), and streptomycin (100 mg/mL) (Gibco). For alkaline phosphatase (ALP) activity, Alizarin Red S staining, real-time PCR, acidic senescence-associated beta-galactosidase (SA-β-Gal) activity assay, and telomere length assay cultures, to prevent the detachment of osteoblasts from the plate during vibration, each well plate was coated with collagen coating using calfskin-derived collagen (Sigma-Aldrich, St. Louis, MO, USA). The collagen acidic solution was diluted 10-fold with 1 mM HCL, added to the plate in the appropriate volume, stretched uniformly, and allowed to stand at room temperature for 1 h, after which the collagen solution was removed. It was then neutralized and dried by three times of washing with DW, and the cell suspension was added to the culture. These cells were cultured at 37 °C and 5% CO2 in calcification medium, adding ascorbic acid (50 mg/mL) and β-glycerophosphate (10 mM) (Cosmo Bio, Tokyo, Japan) to the α-MEM media supplemented with 10% FBS, 1% penicillin G, and 1% streptomycin. All animal experiments were performed in strict accordance with the guidelines of the animal care and use committee of the Tokyo Dental College, Tokyo, Japan. The committee on the ethics of animal experiments of the same college approved the study protocol (permit number: 292898).

### 2.2. Cell Counts

After culturing, cells were washed with Dulbecco’s phosphate buffered saline (DPBS; Gibco) and treated with 0.25% trypsin (Gibco) under standard culture conditions for several minutes. Cultures were then agitated, and the cells were isolated in a Petri dish. The cell suspension was centrifuged for 5 min at 1500 rpm and washed twice with DPBS using the same method. We then measured cell counts in the treated cell suspension using a hemocytometer (Hirschmann Instruments, Eberstadt, Germany).

### 2.3. Vibration System

The vibration was applied using a vibration generator (U56001, 3B scientific, Hamburg, Germany), a function generator (U8533600-115, 3B scientific), and a vibrating plate (U56006, 3B scientific) (Figure 1). The function generator settings were as follows: vibration, sine waves; vibration amplitude, 5V. Well plates were fixated on the vibrating plate, and vibration (3, 30, 300 Hz, or 30, 300 Hz) was applied for 30 min per day to the experimental groups. Control group (0 Hz) specimens were allowed to stand for 30 min at room temperature (25 °C) without vibration.

### 2.4. Cell Proliferation Assay

POB cells were seeded at a density of 1 × 10^4^/well in a 96-well plate (*n* = 8) and cultured for 24 h. Vibration (3, 30, and 300 Hz) was then applied once for 30 min to the experimental groups. After 24 h, 10 μL of WST-1 (Sigma-Aldrich) was added to the experiment and control groups. After 4 h of incubation, the absorbance was measured at 450 nm using a microplate reader (SpectraMax M5e, Molecular Devices, San Jose, CA, USA). The control wavelength was 650 nm.

### 2.5. Cell-Clock Cell Cycle Assay

POB cells were seeded at a density of 1 × 10^5^/well in a 24-well plate (*n* = 3) and cultured for 24 h. Vibrations (3, 30, and 300 Hz) were then applied once for 30 min to the experimental groups. After 24 h, cells in both the experimental and control groups were stained using the Cell-Clock Cell Cycle Assay Kit (Biocolor, Carrickfergus, UK). The cells stained in three colors (yellow, green, and dark blue) were magnified 100-fold upon microscopic imaging. Imaging analysis software ImageJ (NIH, Bethesda, MD, USA) was used to acquire pixel data for each color tone, and cell count proportions were determined for each cell cycle stage based on the proportions of each color tone out of the total pixel values (yellow: G0/G1 stage, green: S/G2 initial stage, dark blue: G2 late stage/M stage).

### 2.6. Evaluation of ALP Activity in Osteoblasts

POB cells were seeded at a density of 1 × 10^5^/well in a 24-well plate (*n* = 3) and cultured until confluence. Cells were then cultured in a calcification medium, and the experimental groups were subjected to vibration (3, 30, and 300 Hz) once every 2 days for 30 min. On day 7, LabAssay ALP (WAKO) was used for assessing ALP activity. We transferred 20 μL of the specimens into a 96-well plate, and the absorbance was measured at 405 nm using a microplate reader. Quantitative evaluation was performed by converting to n-nitrophenol from the calibration curve according to the usual method [12].

### 2.7. Evaluation of Calcification Ability in Osteoblasts

POB cells were seeded at a density of 1 × 10^5^/well in a 24-well plate (*n* = 3) and cultured until confluence. Cells were then cultured in calcification medium, and the experimental groups were subjected to vibration (3, 30, and 300 Hz) once every 2 days for 30 min. On day 14, cells of both groups were washed thrice with DPBS, followed by fixation in 10% formalin. After washing three times with DPBS, cells were stained with 1% Alizarin Red S (Sigma-Aldrich, St. Louis, MO, USA; pH: 5.5), followed by microscopic imaging. A calcified nodule solution (5% formic acid) (Iwai Chemicals, Tokyo, Japan) was used to elute pigments. After 10 min, 100 μL of the eluent was seeded in a 96-well plate, and the absorbance was measured at 425 nm, using a microplate reader [13].

### 2.8. Analysis of mRNA Expression via Real-Time PCR

POB cells were seeded at a density of 5 × 10^5^/well in a 6-well plate (*n* = 3) and cultured until confluence. Cells were then cultured in calcification medium, and the experimental groups were subjected to vibration (30 and 300 Hz) once every 2 days for 30 min. On day 14, total RNA was extracted from the experimental and control groups using TRIzol (Thermo Fisher Scientific, Waltham, MA, USA) and chloroform. After precipitating RNA using isopropanol and washing with 70% ethanol. The extracted total RNA was dissolved in RNase-free water and quantified, and its purity was measured using an ultra-trace spectrometer (Nanodrop 2000, Thermo Fisher Scientific). The ReverTra Ace qPCR RT Master Mix with gDNA Remover (TOYOBO, Osaka, Japan) was used to synthesize cDNA. The THUNDERBIRD SYBR qPCR Mix (TOYOBO) was used for each cDNA for real-time PCR (*n* = 3) in accordance with the SYBR Green method. The PCR conditions were as follows: initial thermal denaturation at 95 °C for 15 min, cycling with thermal denaturation at 95 °C for 3 s, annealing at 62 °C for 5 s, and elongation at 72 °C for 45 s for 40 cycles. Targets of mRNA expression analysis were runt-related transcription factor 2 (Runx2), bone morphogenetic protein 2 (BMP2), collagen type I alpha 1 chain (Col1a1), osteopontin (OPN), osteocalcin (OCN), and sclerostin, all being osteoblast differentiation markers. We also targeted p53, p21, and p16, aging-related markers, and c-fos, a cytotoxicity marker. Primers are shown in Table 1. With beta-actin as the internal control, relative mRNA expression was quantified using the double delta method.

### 2.9. Evaluation of SA-β-Gal Activity for Cell Aging

POB cells were seeded at a density of 1 × 10^4^/well in a 96-well plate (*n* = 5) and cultured until confluence. Cells were then cultured in calcification medium, and the experimental groups were subjected to vibration (30 and 300 Hz) once every 2 days for 30 min. On day 14, a 96-well cellular senescence assay kit (Cell Biolabs, San Diego, CA, USA) was used, and fluorometry was performed for the experimental and control groups with an excitation wavelength of 360 nm and an emission wavelength of 465 nm, using a microplate reader. Total protein was quantified using a BCA Kit (Thermo Fisher Scientific), and the results of SA-β-Gal measurement were normalized.

### 2.10. Telomere Length Assay (qPCR)

POB cells were seeded at a density of 5 × 10^5^/well in a 6-well plate (*n* = 3) and cultured until confluence. Cells were then cultured in calcification medium, and the experimental groups were subjected to vibration (30 and 300 Hz) once every 2 days for 30 min. On day 14, a High Pure PCR Template Preparation Kit (Roche Diagnostics, Mannheim, Germany) was used to extract DNA, followed by quantification and purity assessment using an ultra-trace spectrometer. Telomere length was assessed using the Absolute Mouse Telomere Length Quantification qPCR Assay Kit (AMTLQ) (ScienCell, Carlsbad, CA, USA). The PCR conditions were as follows: initial thermal denaturation at 95 °C for 10 min, cycling with thermal denaturation at 95 °C for 20 s, and annealing at 52 °C for 20 s, elongation at 72 °C for 45 s for 32 cycles. With the Reference Mouse genomic DNA sample (telomere length: 3.79 ± 0.23 Mb per diploid cell) as the control, quantification was performed using the double delta method.

### 2.11. Statistical Analysis

For each experiment, excluding the Cell-Clock Cell Cycle Assay, multiple comparisons were made among the experimental groups (3, 30, and 300 Hz) and the control group (0 Hz), using one-way ANOVA, followed by Dunnet’s test for post hoc analysis. Tests were performed using IBM SPSS Statics for Windows (Ver. 24.0, IBM, Armonk, NY, USA). A *p*-value less than 0.05 was considered to indicate statistical significance (*: *p* < 0.05, **: *p* < 0.01).

## 3. Results

### 3.1. Effects of Vibration on Cell Proliferation

The cell proliferation assay revealed that vibration at 30 and 300 Hz yielded a significant increase in cell proliferation compared to the control group (*p* < 0.01); however, no significant differences were observed at 3 Hz (Figure 2).

### 3.2. Effects of Vibration on the Cell Cycle

The Cell-Clock Cell Cycle Assay revealed that at the control group and 3 Hz vibrations, fewer differences were observed in each color (Figure 3A,B,E). At 30 Hz, more significant values of blue-stained cells and smaller values of yellow-stained cells were observed compared to the control group (Figure 3A,C,E). On applying vibrations at 300 Hz, a smaller number of green-stained cells and a larger number of yellow-stained cells were observed (Figure 3A,D,E).

### 3.3. Vibration Accelerates Osteoblast Differentiation

Compared to the control group, the ALP activity was significantly higher in the group exposed to 30 Hz vibrations (*p* < 0.01); however, no significant differences were observed in cells exposed to 3 and 300 Hz compared to the control group (Figure 4A). Upon Alizarin Red staining, calcium deposition increased significantly in cells exposed to 30 Hz vibrations as compared to the control group (*p* < 0.01), as evident from an increase in the staining intensity (Figure 4B,C); however, no significant differences were observed in cells exposed to 3 and 300 Hz vibrations as compared to the control group.

### 3.4. Vibrations Cause Changes in Osteoblast Differentiation Marker Genes

Real-time PCR analysis indicated that Runx2 mRNA was significantly downregulated in cells exposed to 30 Hz and 300 Hz vibrations compared to that in the control group (30 Hz, *p* < 0.01; 300 Hz, *p* < 0.05). BMP2 mRNA expression did not differ significantly; however, it displayed a decreasing trend at 30 and 300 Hz. Furthermore, OPN and OCN mRNAs, being middle-to-late-stage differentiation markers of osteoblasts, sclerostin mRNA, which is generally upregulated in osteocytes, and Col1a1 mRNA, which is generally expressed throughout differentiation, were significantly upregulated in the 30 Hz group (OPN, *p* < 0.01; OCN, *p* < 0.05; Sclerostin, *p* < 0.05; Col1a1, *p* < 0.01) (Figure 5).

### 3.5. Vibration Accelerates Cellular Aging

Upon analysis of cellular aging, p53/p21 and p16 mRNA, markers of cellular aging, were significantly upregulated in the 30 Hz group (*p* < 0.01) (Figure 6A). Moreover, c-fos, a cellular stress marker, was significantly upregulated only in the 30 Hz group (*p* < 0.05) (Figure 6B). SA-β-Gal, a cellular aging marker, displayed significantly increased activity in the cells exposed to 30 Hz and 300 Hz compared to cells in the control group (Figure 6C) (*p* < 0.01). Telomere length was significantly shorter in the 30 Hz (4.02 ± 0.24 Mb) and 300 Hz (3.87 ± 0.23 Mb) groups than in the control group (7.36 ± 0.46 Mb) (*p* < 0.01) (Figure 6D).

## 4. Discussion

This study aimed to investigate the effect of vibration on cell cycle progression, proliferation, differentiation, and aging in primary osteoblasts harvested from the maxilla of eight-week-old mice rather than in cell lines such as MC3T3-E1 cells. We isolated osteoblasts via the enzymatic method [14,15,16,17] and report that the isolated cells were osteoblasts, evident from calcification and osteoblast-specific mRNA expression.

In the present study, vibrations at 3, 30, and 300 Hz were applied to the experimental groups. Vibration at 30 Hz was referenced from previous studies of the vibrator for orthodontic treatment (10,11), and vibration at 3 and 300 Hz (one-tenth and 10 times of 30 Hz, respectively) was adopted for comparison with 30 Hz. Moreover, to prevent cells from floating, a collagen coating was applied to the well plates, and the vibration was applied once every 2 days for 30 min; in previous studies, the vibration was applied once per day for 30 min.

In an in vitro experimental system, osteoblasts proliferated until confluence, followed by differentiation. Accordingly, it is appropriate to evaluate cells until confluence when comparing cell proliferation and cell cycles upon the application of vibration. Moreover, to assess cell differentiation and aging, the culture medium needs to be changed to a calcification medium once the cells are confluent. Therefore, in the present study, we evaluated cell proliferation and cell cycles 24 h after vibration. We assessed cell differentiation and aging on days 7 or 14 after switching the medium to a calcification medium at the confluence, since each ALP activity and calcium deposition increases on day 7 and day 14 [7,8].

The cell proliferation assay revealed that in the 30 Hz group, cell proliferation increased significantly soon after vibration. The Cell-Clock Cell Cycle Assay revealed that oxidation and reduction states change upon cell cycle progression, as reflected through changes in the color tone of the redox dye [18]. In the present study, numerous cells from the latter half of the cell cycle were observed at 30 Hz, suggesting that vibration accelerates the cell cycle.

The osteoblastic differentiation process could be divided into several stages, including proliferation, extracellular matrix deposition, matrix maturation, and mineralization [19]. Known studies have shown that vibration is effective in cell proliferation [20,21]. Differentiated osteoblasts cease to proliferate and start producing and secreting the characteristic extracellular matrix of bone tissue, accompanied by calcification [22,23,24,25]. The differentiation and maturation process can be surmised on the basis of osteoblast differentiation markers. ALP staining and Alizarin Red staining revealed that mid- to long-term (1–2 weeks) application of vibration significantly increased ALP activity and calcium deposition at 30 Hz, suggesting that vibration increases the secretion of extracellular matrix proteins in osteoblasts and promotes the deposition of calcified bodies. However, in this study, collagen coating was applied to prevent osteoblasts from being detached by vibration, and since the arginine-glycine-aspartic acid (RGD) amino acid sequence in collagen promotes cell growth, adhesion, and interaction with integrin, we believe that its influence on cell behavior cannot be ruled out [26]. Here, real-time PCR was used to evaluate late differentiation markers in the osteoblasts on day 14 after replacement with a calcification medium to determine if the osteoblasts were heading toward calcification. In addition, while OPN is considered to have peaks in proliferation and late differentiation, here we used it as an indicator of late osteoblast differentiation [27]. Real-time PCR analysis indicated that on applying vibrations to osteoblasts, Runx2 and BMP2 mRNA, early-stage differentiation markers for osteoblast differentiation, were significantly downregulated or displayed a decreasing trend at 30 Hz compared to the control group. OPN and OCN, late-stage differentiation markers, and sclerostin, an osteocyte differentiation marker, were upregulated at 30 Hz compared to the control group, suggesting that vibration accelerates the differentiation of undifferentiated MSCs into osteoblasts and subsequently into osteocytes.

Regarding aging, at the genetic level, vibration at 30 Hz significantly upregulated p53, p21, and p16. At the protein level, SA-β-Gal, an aging marker, displayed significantly increased activity. Furthermore, telomere length was significantly decreased at 30 Hz (54% length compared to that at 0 Hz). p53 is activated upon telomere shortening and DNA damage resulting from various types of stress, resulting in cell cycle arrest, apoptosis, DNA repair, and inhibition of cancer cell metastasis and angiogenesis [28]. Moreover, mice harboring a hyperactive p53 reportedly have lower cancer onset rates but faster tissue aging and shorter lifespans [29]. Activated p53 also activates p21, a CDK inhibitor in the cell cycle, thereby causing a cell cycle arrest. In addition, p16, a CDK4 and CDK6 inhibitor in the cell cycle, is a cell aging marker closely associated with the aging characteristics [30,31].

SA-β-Gal is located in lysosomes, and its optimal pH tends towards the acidic range. It is an enzyme and an aging marker that has received increasing attention owing to its increased activity at a near-neutral pH in senescent cells [32,33,34]. The cellular senescence pathway can be divided into telomere-dependent aging caused by cell division and telomere-independent aging caused by various stresses. In the present study, telomere length was shortened; hence, cells were considered to undergo telomere-dependent aging.

Previous studies have reported that the application of vibration to osteoblasts upregulates genes, making it necessary for cells to stretch by approximately 10% of their size [35]. Other studies, however, reported that upon application of vertical vibration, differences in nuclear and cytoplasmic movement are more critical than a stretch of cell size [36,37,38]. In the present study, we used vertical vibrations and found that vibration at 30 Hz was effective for osteoblasts. No significant differences were observed at 3 Hz, while vibration at 300 Hz significantly increased cell proliferation, SA-β-Gal activity, the expression of some genes upon real-time PCR, and significantly decreased telomere length in comparison with the control group (0 Hz). No significant differences were observed in several experiments, suggesting that the vibration at 3 and 300 Hz may not have induced sufficient differences in nuclear and cytoplasmic movement. Furthermore, unmatched results between several experiments were observed at 300 Hz. Further studies are required to elucidate the conflicting results of these experiments.

Real-time PCR analysis revealed that c-fos, a known marker of osteoblast proliferation and differentiation [39,40], was significantly upregulated at 30 Hz alone, suggesting that in the present system, 30 Hz vibration induced continuous changes in mRNA and protein levels. These results suggest that application of optimal vibration to osteoblasts promotes cell proliferation by accelerating the cell cycle and that expression of cytotoxicity markers, such as c-fos, causes calcification by accelerating cell differentiation upon application of mid- to long-term vibration. Moreover, the aging phenomenon was seemingly induced in a telomere-dependent manner.

Vibration is currently being applied in various fields of dentistry. For example, in orthodontic research, vibration is being applied to accelerate tooth movement [41]. In addition to the effects of vibration on osteoblasts, a recent study investigated its effects on osteoclasts [35]. Furthermore, studies are required to examine not only the anabolic effects of vibration but also its catabolic effects. In this study, I believe that vibration promotes osteoblast differentiation and leads to mineralization. Because orthodontic tooth movement (OTM) requires bone resorption on the compression side and bone addition on the traction side, increased bone addition on the traction side facilitates tooth movement. However, accelerated osteoblast differentiation on the compression side may disadvantage tooth movement. Since this is an in vitro study, detailed in vivo experiments will be needed to see if it can be applied in clinical settings with complex intercellular linkages. Moreover, if further studies elucidate the effects of vibration on osteogenesis and osteoclastogenesis in the tooth root area within the inflammatory environment, acceleration of tooth movement can also be established. In the field of oral implantology, initial fixation [42] for diabetes, bone grafts, and osteoporosis is currently a problem [43,44], and clinical applications may be possible if a method to control cell proliferation, differentiation, and calcification by vibration can be clarified. Such improvements in healing and differentiation can lead to better results for both soft and hard tissues around the implants [45,46].

Previous studies have reported an increase in calcification and upregulated mRNA expression of osteoblast differentiation markers after vibration. However, few studies have investigated the expression of sclerostin and changes of aging-related markers such as p53, p21, p16, SA-β-Gal, and Telomere length, considered as temporal indicators. In the present study, an increase of sclerostin and aging-related markers after vibration suggests that vibration accelerates the differentiation of osteoblasts into osteocytes and induces cellular aging. Thus, the present study clarified for the first time that vibration applied to osteoblast-induced cell aging. However, since vibration enhances cell proliferation and calcification, the application of vibration to dental treatment is valuable. Meanwhile, worthy of clinical application, an attempt should be made to observe the mobility of MSCs or osteoblasts to the alveolar bone areas from distant bone marrow via blood circulation in vivo, given that long-term vibration results in aging and subsequently in osteoblast depletion.

## 5. Conclusions

This study shows that vibrations at 30 Hz resulted in significant differences in cell cycle and cell proliferation. The cell cycle was accelerated, and cell proliferation was induced immediately after application of optimal vibration to osteoblasts. In the mid- to long-term stage after application of optimal vibration to osteoblasts, cell differentiation and calcification were promoted through an increase in c-fos expression. Upon application of optimal vibration to osteoblasts, cellular aging was accelerated through an increase in cell division, resulting from the acceleration of the cell cycle and telomere shortening. Controlled vibration inducing osteogenesis could be applied during clinical orthodontic treatment, upon initial fixation with an implant anchor, or during the retention period.

## Figures and Tables

**Figure 1 biomedicines-11-00444-f001:**
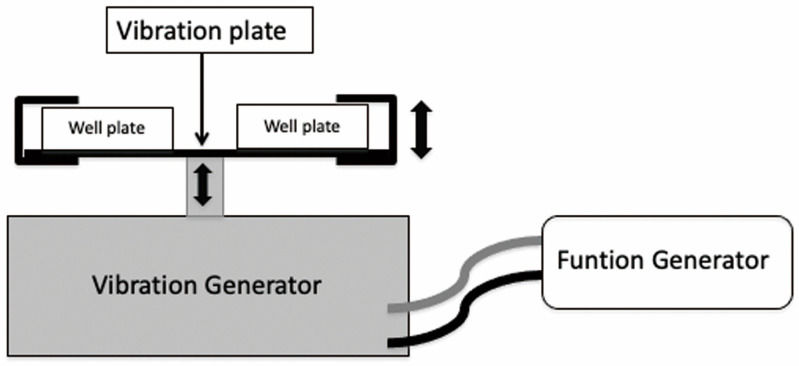
The vibration system used in the study.

**Figure 2 biomedicines-11-00444-f002:**
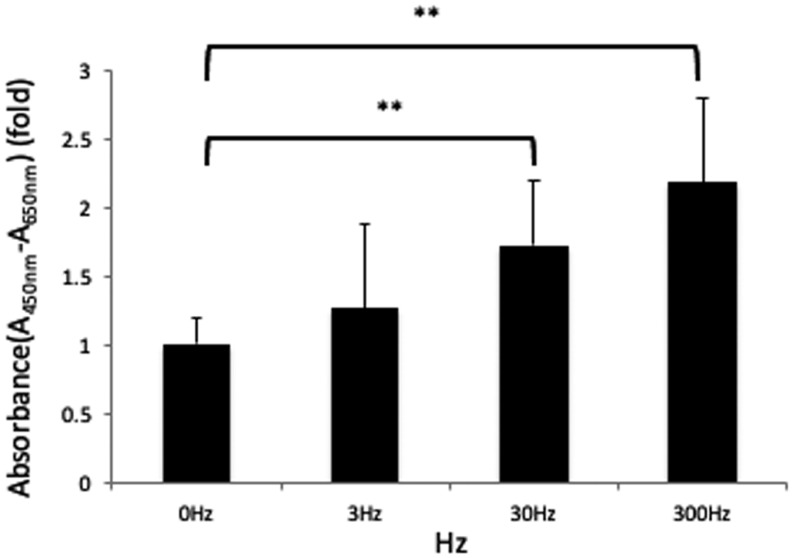
Cell Proliferation Assay. At 24 h after the application of vibration at 0, 3, 30, or 300 Hz, cells were evaluated with WST-1. Cell proliferation increased significantly at 30 and 300 Hz rather than at 0 Hz. Dunnet’s multiple comparison test for 30 Hz and 300 Hz to 0 Hz (**: *p* < 0.01).

**Figure 3 biomedicines-11-00444-f003:**
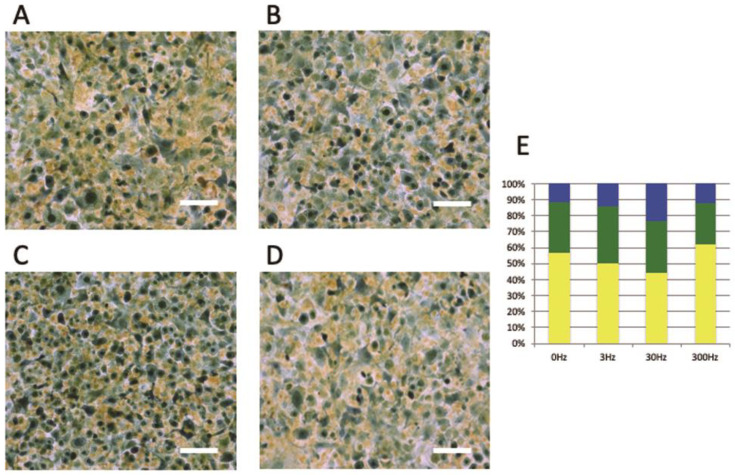
Cell-Clock Cell Cycle Assay. (**A**): 0 Hz, (**B**): 3 Hz, (**C**): 30 Hz, (**D**): 300 Hz, Scale bar = 100 µm, (**E**): Each cell cycle stage is based on the proportions of each color tone out of the total pixel values. Many cells were stained dark blue upon applying vibrations at 30 Hz compared to 0, 3, and 300 Hz. (yellow: G0/G1 stage, green: S/G2 initial stage, dark blue: G2 late stage/M stage).

**Figure 4 biomedicines-11-00444-f004:**
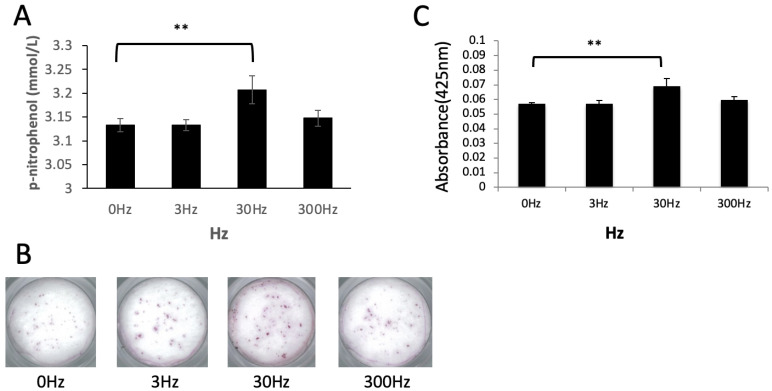
**ALP activity and Alizarin Red staining**. (**A**): Alkaline phosphatase (ALP) activity, (**B**): Alizarin Red staining, (**C**): Alizarin Red staining intensity. In (**A**,**C**), ALP activity and calcium deposition were significantly increased at 30 Hz rather than at 0 Hz. Dunnet’s multiple comparison test for 30 Hz and 300 Hz to 0 Hz (**: *p* < 0.01).

**Figure 5 biomedicines-11-00444-f005:**
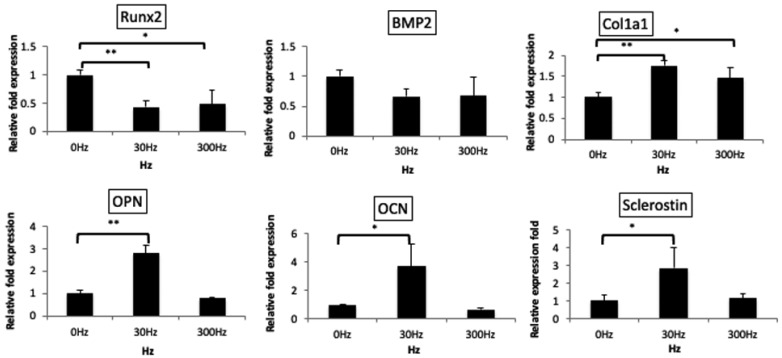
**mRNA expression of differentiation markers**. Early-stage differentiation markers were downregulated, while later-stage markers were significantly upregulated at 30 Hz rather than at 0 Hz. Dunnet’s multiple comparison test for 30 Hz and 300 Hz to 0 Hz (*: *p* < 0.05, **: *p* < 0.01).

**Figure 6 biomedicines-11-00444-f006:**
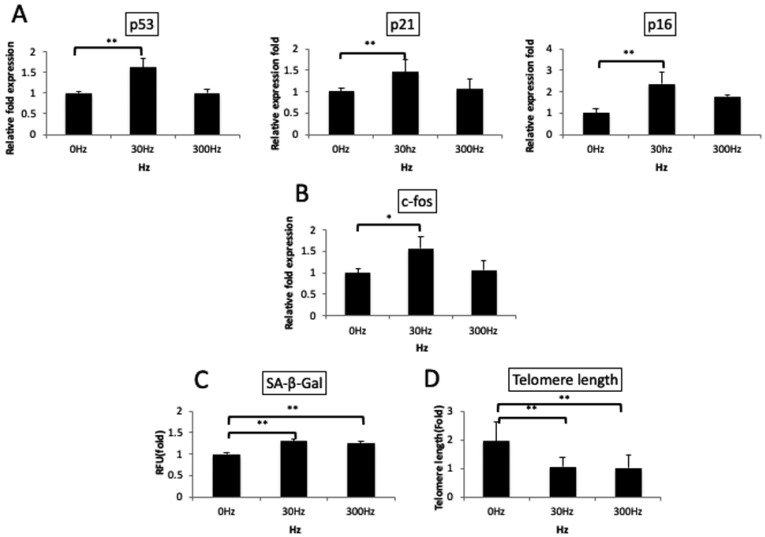
**Cellular aging resulting from the vibration of osteoblasts.** (**A**,**B**): Real-time PCR, (**C**): Senescence-associated beta-galactosidase (SA-β-Gal) assay, (**D**): Telomere length assay (qPCR) (**A**): On applying vibrations at 30 Hz, aging-related genes (p53, p21, and p16) were significantly upregulated. (**B**): A stress-related gene (c-fos) was significantly upregulated at 30 Hz. (**C**): SA-β-Gal activity, an aging marker, was significantly upregulated at 30 Hz and 300 Hz. (**D**): Telomere length was significantly reduced at 30 Hz and 300 Hz. Dunnet’s multiple comparison test for 30 Hz and 300 Hz to 0 Hz (*: *p* < 0.05, **: *p* < 0.01).

**Table 1 biomedicines-11-00444-t001:** Design of primers.

Gene	Forward (5′-3′)	Reverse (5′-3′)
β-actin	CGGTTCCGATGCCCTGAGGCTCTT	CGTCACACTTCATGATGGAATTGA
Runx2	AATTAACGCCAGTCGGAGCA	CACTTCTCGGTCTGACGACG
BMP2	CTCTCTCAATGGACGTGCCC	AACACTAGAAGACAGCGGGTC
Col1a1	TTCTCCTGGCAAAGACGGAC	CGGCCACCATCTTGAGACTT
OPN	GGCTGAATTCTGAGGGACTAACT	ACAGCATTCTGTGGCGCAAG
OCN	GTATGGCTTGAAGACCGCCTA	AGGGAGGATCAAGTCCCG
Sclerostin	GCCTTCAGGAATGATGCCAC	CTGTCAGGAAGCGGGTGTAG
p53	AAACGCTTCGAGATGTTCCG	CAAGGCTTGGAAGGCTCTAGG
p21	TAAGGACGTCCCACTTTGCC	AAAGTTCCACCGTTCTCGGG
p16	CGAACTCGAGGAGAGCCATC	TACGTGAACGTTGCCCATCA
c-fos	TACTACCATTCCCCAGCCGA	GCTGTCACCGTGGGGATAAA

## Data Availability

Not applicable.

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
