# Peer review of "Vertical Vibration of Mouse Osteoblasts Promotes Cellular Differentiation and Cell Cycle Progression and Induces Aging In Vitro"

_biomedicines, 2023, doi:10.3390/biomedicines11020444_

Round 1

Reviewer 1 Report

The authors aimed to investigate the effect of vibration of osteoblasts on the cell cycle, cell differentiation, and aging.

It ought to be specified that only vertical vibrations were tested earlier in the manuscript (now mentioned in line 335), or perhaps in the title, as horizontal and vertical vibration have different effect on  cells

Line 66-69 … a collagen coating was provided with collagen derived from calf skin (Sigma-Aldrich, St. Louis, MO) for each well plate.

This was apparently to avoid floating of the cells, according to information provided in the discussion part. It can only be assumes that this is collagen type I, but at what concentration and/or of which thickness was this applied? Was this applied as a coating on top of the cells, or as a coating in the wells prior to culturing the cells?  Collagen has clear effects on the cells alone, why is this not discussed?

 Why was only one time point measured for ALP (d 7), Alizarin red (d 14), PCR analyses of osteoblast markers and ageing (d 14), respectively? Osteoblastogenesis involves three major stages and the markers peaks at different stages during differentiation. OPN is identified to be middle-to-late-stage differentiation marker of osteoblasts in this manuscript (line 247), however OPN peaks twice, during proliferation and then again in the later stages of differentiation (Huang et al. 2017  doi: 10.2741/2296).

There is a separate section on cell counts (2.2). Is this section referring to a separate cellular set up, or were cells counted after all the experiments? Data are apparently not presented.

In the cell-clock cell cycle assay (2.5) a density of 1×105 cells/well in a 24-well plate were used, and harvested after 24 hours (Figure 3),  whereas in the Alizarin red measurements, the same cell density was applied in 24 well plates, cells were cultured until confluence, than with calcification medium and harvested at day 14 (Figure 4B). The quality of figure 4B is low, but it appears to have less cell density here than in the figure 3

Related to figure 4 A and B, why are absorptions values presented and not values normalized to a calibration curves?

Line 81.  Cultures were then agitatedis agitated the right description?

The discussion part needs to be rewritten. It is repeating the results without discussing these, and several places references are lacking after statements, e.g line 376-77 – the previous studies are not referred to.  

 Line 298 Undifferentiated MSCs differentiate into osteoblast precursors, pre-osteoblasts, osteoblasts, and osteocytes before finally undergoing apoptosis [17].

This is not a previse presentation of the differentiation stages from MScs to osteocytes.  In the referred article, they are investigating factors involved in osteoblast differentiation into osteocytes and apoptosis – in other words osteoblasts undergo either apoptosis (60-65 %) or differentiation into osteocytes (or lining cells).  Besides being cited wrongly, this reference nr 17 does not mention MSCs, pre-osteoblasts etc either.

 Line 304-306 increased ALP activity and calcium deposition at 30 Hz, suggesting that vibration increases the secretion of extracellular matrix proteins in osteoblasts and promotes the deposition of calcified bodies.

Extra cellular matrix was not evaluated.  A collagen coating was applied, and the influence of this on cellular behavior cannot be excluded and thus the interpretation of the results and the conclusion in this paper; e.g. the arginine-glycine-aspartate (RGD) amino acid sequence in collagen can promote cell growth, attachment, and interact with integrin [Ferraro et al. Int. J. Biol. Macromol 2017; 97:55 – 66].

 It is not clear why the section from line 365-374 is included in the discussion part. The relevance to the present study is lacking.

Author Response

Reviewer 1

We sincerely appreciate your taking the time to review this paper in detail.We have made all revisions to the areas you have pointed out in accordance with your suggestions. As a result, many improvements have been made compared to the first edition. Our thanks again to you.

#1: It ought to be specified that only vertical vibrations were tested earlier in the manuscript (now mentioned in line 335), or perhaps in the title, as horizontal and vertical vibration have different effect on cells

A1: I have added "Vertical" to the title as you suggested.

#2: Line 66-69 … a collagen coating was provided with collagen derived from calf skin (Sigma-Aldrich, St. Louis, MO) for each well plate.This was apparently to avoid floating of the cells, according to information provided in the discussion part. It can only be assumes that this is collagen type I, but at what concentration and/or of which thickness was this applied? Was this applied as a coating on top of the cells, or as a coating in the wells prior to culturing the cells?  Collagen has clear effects on the cells alone, why is this not discussed?

A2: In "Materials and Methods 2.1 cell culture," the following has been added: "To keep osteoblasts attached to the plate from detaching due to vibration, each well plate was coated with collagen coating using calfskin-derived collagen (Sigma-Aldrich, St. Louis, MO). The collagen acidic solution was diluted 10-fold with 1 mM HCL, added to the plate in the appropriate volume, stretched uniformly, and allowed to stand at room temperature for 1 hour, after which the collagen solution was removed. It was then neutralized and dried by three times of washing with DW, and the cell suspension was added to the culture."

#3: Why was only one time point measured for ALP (d 7), Alizarin red (d 14), PCR analyses of osteoblast markers and ageing (d 14), respectively? Osteoblastogenesis involves three major stages and the markers peaks at different stages during differentiation. OPN is identified to be middle-to-late-stage differentiation marker of osteoblasts in this manuscript (line 247), however OPN peaks twice, during proliferation and then again in the later stages of differentiation (Huang et al. 2017 doi: 10.2741/2296).

A3: The following revision and references have been added: " The osteoblastic differentiation process could be divided into several stages, including proliferation, extracellular matrix deposition, matrix maturation, and mineralization [19]. Known studies have shown that vibration is effective in cell proliferation [20,21]. Here, real-time PCR was used to evaluated late differentiation markers in the osteoblasts on day 14 after replacement with a calcification medium to determine if the osteoblasts were heading toward calcification. In addition, while OPN is considered to have peaks in proliferation and late differentiation, here we used it as an indicator of late osteoblast differentiation [27]."

#4: There is a separate section on cell counts (2.2). Is this section referring to a separate cellular set up, or were cells counted after all the experiments? Data are apparently not presented.

A4: As you noted, we have removed this section as it is unnecessary for this section.

#5: In the cell-clock cell cycle assay (2.5) a density of 1×105 cells/well in a 24-well plate were used, and harvested after 24 hours (Figure 3), whereas in the Alizarin red measurements, the same cell density was applied in 24 well plates, cells were cultured until confluence, than with calcification medium and harvested at day 14 (Figure 4B). The quality of figure 4B is low, but it appears to have less cell density here than in the figure 3.

A5: In the 14 days of culturing osteoblasts in the calcification medium, few calcified nodules and few areas stained with alizarin red S are expected. Figure 4B shows that the addition of 30 Hz vibration results in more calcification than under other conditions. Therefore, it is believed that the number of cells appears to be low since only the calcification is stained, not the cells themselves. The cell count was determined to be correct as described.

#6: Related to figure 4 A and B, why are absorptions values presented and not values normalized to a calibration curves?

A6: Regarding the ALP activity assay, the graph and text have been revised as follows: "We transferred 20 µl of specimen to 96 well plates and measured the absorbance at 405 nm using a microplate reader. The graph was modified by converting from calibration curve to n-nitrophenol." Also, reference [12] was added. For alizarin red, since it is common to express it using absorbance, reference [13], which reports a similar method, was added.

#7: Line 81.  Cultures were then agitated – is agitated the right description?

A7: As you indicated, this is a mistake and has been removed.

#8: The discussion part needs to be rewritten. It is repeating the results without discussing these, and several places references are lacking after statements, e.g line 376-77 – the previous studies are not referred to.

A8: As you pointed out, we have deleted it because it lacks evidence. We have also removed similar content from the discussion section.

#9: Line 298 Undifferentiated MSCs differentiate into osteoblast precursors, pre-osteoblasts, osteoblasts, and osteocytes before finally undergoing apoptosis [17].This is not a previse presentation of the differentiation stages from MScs to osteocytes.  In the referred article, they are investigating factors involved in osteoblast differentiation into osteocytes and apoptosis – in other words osteoblasts undergo either apoptosis (60-65 %) or differentiation into osteocytes (or lining cells).  Besides being cited wrongly, this reference nr 17 does not mention MSCs, pre-osteoblasts etc either.

A9: As you indicated, we found [17] to be inappropriate. We have revised the text by adding appropriate references.” The osteoblastic differentiation process could be divided into several stages, including proliferation, extracellular matrix deposition, matrix maturation, and mineralization [19].”

#10:  Line 304-306 increased ALP activity and calcium deposition at 30 Hz, suggesting that vibration increases the secretion of extracellular matrix proteins in osteoblasts and promotes the deposition of calcified bodies. Extra cellular matrix was not evaluated.  A collagen coating was applied, and the influence of this on cellular behavior cannot be excluded and thus the interpretation of the results and the conclusion in this paper; e.g. the arginine-glycine-aspartate (RGD) amino acid sequence in collagen can promote cell growth, attachment, and interact with integrin [Ferraro et al. Int. J. Biol. Macromol 2017; 97:55 – 66].

A10: As you have pointed out, since we did not analyze the extracellular matrix, we revised the text in this section, added references, and connected it to the discussion of real-time PCR. "ALP staining and alizarin red staining revealed that a vibration of 30 Hz over a medium- to long-term period (1-2 weeks) significantly increased ALP activity and calcium deposition, suggesting that vibration increases the cellular activity of osteoblasts and promotes deposition of calcified bodies. However, in this study, collagen coating was applied to prevent osteoblasts from being detached by vibration, and since the arginine-glycine-aspartic acid (RGD) amino acid sequence in collagen promotes cell growth, adhesion, and interaction with integrin, we believe that its influence on cell behavior cannot be ruled out [26]."

#11:  It is not clear why the section from line 365-374 is included in the discussion part. The relevance to the present study is lacking.

A11: We considered deleting the part of Line 365-374, but Reviewer 2 instructed us to add more details. However, we understand your point and after due consideration, we have revised the text regarding the dental application of vibration and added references as directed by Reviewer 2. “Vibration is currently being applied in various areas of dentistry. For example, in orthodontic research, the promotion of tooth movement via vibration is being considered. Furthermore, it is necessary to study not only the anabolic but also the catabolic effects of vibration [41]. In the field of orthodontic treatment and oral implantology the initial fixation [42] for diabetes, bone grafts, and osteoporosis is currently a problem [43, 44], and clinical applications may be possible if a method to control cell proliferation, differentiation, and calcification by vibration can be clarified. Such improvement in healing and differentiation can lead to better results for both soft than hard tissues around the implants [45,46].”

Reviewer 2 Report

The paper is well written and very interesting, in each clinical situation where bone differentiation is required. In particular, the referral to the field of oral implantology is mandatory, where, a faster bone differentation, will bring to important reduction of time of treatment for patients. 

So, it is necessary to improve the paper with more references to the oral implantology field, in particular:

1) Such vibration, will improve osseintegration in different situations, like bone grafting or diabetes patients? Please cite PubMed ID36142007and DOI10.3390/app12136729

2) Such improvement in healing and differentiation can lead to better results for both soft than hard tissues around the implants. cite DOI10.23805/JO.2018.10.04.04

Author Response

Reviewer 2

We sincerely appreciate your taking the time to review this paper in detail.We have made all revisions to the areas you have pointed out in accordance with your suggestions. As a result, many improvements have been made compared to the first edition. Our thanks again to you.

So, it is necessary to improve the paper with more references to the oral implantology field, in particular:

Regarding the application in the field of dentistry, we have added content from the field of oral implantology to our discussion, as you have indicated.

1) Such vibration, will improve osseintegration in different situations, like bone grafting or diabetes patients? Please cite PubMed ID36142007and DOI10.3390/app12136729

A1: We have revised our discussion by adding the references you mentioned. “Vibration is currently being applied in various areas of dentistry. For example, in orthodontic research, the promotion of tooth movement via vibration is being considered. Furthermore, it is necessary to study not only the anabolic but also the catabolic effects of vibration [41]. In the field of orthodontic treatment and oral implantology the initial fixation [42] for diabetes, bone grafts, and osteoporosis is currently a problem [43, 44], and clinical applications may be possible if a method to control cell proliferation, differentiation, and calcification by vibration can be clarified.”

2) Such improvement in healing and differentiation can lead to better results for both soft than hard tissues around the implants. cite DOI10.23805/JO.2018.10.04.04

A2: We have revised our discussion by adding the references you mentioned. “Such improvement in healing and differentiation can lead to better results for both soft than hard tissues around the implants [45,46].”

Round 2

Reviewer 1 Report

The authors have taken into account previous comments, but there is a contradiction in the revised version of the manuscript. The authors find that vibration increases cell differentiation and mineralization, while writing in the discussion section that vibration is used clinically to accelerate the rate of tooth movement (OTM rate) (line 371-2). Increased mineralization results in higher bone density and stronger bones, and thus a reduced, not accelerated, tooth movement is to be expected. This shows that it is difficult to extrapolate results based on single cell experiments to a complex in vivo setting.

Lines 381-391 are not well written, and it is difficult to follow the arguments

Author Response

Dear Reviewer 1

I am very grateful for the opportunity to revise this paper again. The following is my response to the reviewer regarding the correction. Thanks to the detailed peer review, I believe I have improved the content to make it easier for readers to understand not only the biomechanisms but also the actual clinical application. I am sorry to take your time, but I would appreciate it if you could review it again.

Q1: The authors have taken into account previous comments, but there is a contradiction in the revised version of the manuscript. The authors find that vibration increases cell differentiation and mineralization, while writing in the discussion section that vibration is used clinically to accelerate the rate of tooth movement (OTM rate) (line 371-2). Increased mineralization results in higher bone density and stronger bones, and thus a reduced, not accelerated, tooth movement is to be expected. This shows that it is difficult to extrapolate results based on single cell experiments to a complex in vivo setting.

A1: I have corrected and added the sentence as above.

“Vibration is currently being applied in various fields in dentistry. For example, in orthodontic research, vibration is being applied to accelerate tooth movement [41]. In addition to the effects of vibration on osteoblasts, a recent study investigated its effects on osteoclasts [35]. Furthermore, studies are required to examine not only the anabolic effects of vibration but also its catabolic effects. In this study, I believe that vibration promotes osteoblast differentiation and leads to mineralization. Because orthodontic tooth movement (OTM) requires bone resorption on the compression side and bone addition on the traction side, increased bone addition on the traction side facilitates tooth movement. However, accelerated osteoblast differentiation on the compression side may disadvantage tooth movement. Since this is an in vitro study, detailed in vivo experiments will be needed to see if it can be applied in clinical settings with complex intercellular linkages. Moreover, if further studies elucidate the effects of vibration to osteogenesis and osteoclastogenesis in tooth root area within the inflammatory environment, acceleration of tooth movement also can be established.” 

Q2: Lines 381-391 are not well written, and it is difficult to follow the arguments

A2: I have corrected and added the sentence as above.

“In the field of oral implantology the initial fixation [42] for diabetes, bone grafts, and osteoporosis is currently a problem [43, 44], and clinical applications may be possible if a method to control cell proliferation, differentiation, and calcification by vibration can be clarified. Such improvement in healing and differentiation can lead to better results for both soft than hard tissues around the implants [45,46].”